# Airway Management: The Current Role of Videolaryngoscopy

**DOI:** 10.3390/jpm13091327

**Published:** 2023-08-29

**Authors:** Sophie A. Saul, Patrick A. Ward, Alistair F. McNarry

**Affiliations:** 1St. John’s Hospital, Howden West Road, NHS Lothian, Livingston EH54 6PP, UK; sophie.saul@nhs.scot (S.A.S.); althegasman@btinternet.com (A.F.M.); 2Western General Hospital, Crewe Road South, NHS Lothian, Edinburgh EH4 2XU, UK

**Keywords:** videolaryngoscopy, videolaryngoscope, laryngoscopy, airway management

## Abstract

Airway management is usually an uncomplicated and safe intervention; however, when problems arise with the primary airway technique, the clinical situation can rapidly deteriorate, resulting in significant patient harm. Videolaryngoscopy has been shown to improve patient outcomes when compared with direct laryngoscopy, including improved first-pass success at tracheal intubation, reduced difficult laryngeal views, reduced oxygen desaturation, reduced airway trauma, and improved recognition of oesophageal intubation. The shared view that videolaryngoscopy affords may also facilitate superior teaching, training, and multidisciplinary team performance. As such, its recommended role in airway management has evolved from occasional use as a rescue device (when direct laryngoscopy fails) to a first-intention technique that should be incorporated into routine clinical practice, and this is reflected in recently updated guidelines from a number of international airway societies. However, currently, overall videolaryngoscopy usage is not commensurate with its now widespread availability. A number of factors exist that may be preventing its full adoption, including perceived financial costs, inadequacy of education and training, challenges in achieving deliverable decontamination processes, concerns over sustainability, fears over “de-skilling” at direct laryngoscopy, and perceived limitations of videolaryngoscopes. This article reviews the most up-to-date evidence supporting videolaryngoscopy, explores its current scope of utilisation (including specialist techniques), the potential barriers preventing its full adoption, and areas for future advancement and research.

## 1. Introduction

Tracheal intubation is the gold standard in airway management, permitting both positive pressure ventilation and protection from pulmonary aspiration. Tools to view the larynx have been described since the 18th century, although Chevalier Jackson is credited as the first clinician to combine direct visualisation of the larynx with passage of a tracheal tube through the vocal cords in 1913 [1]. Various iterations of the direct laryngoscope followed (so called as they are used to achieve a direct line of sight between the laryngoscopist’s eye and the glottis) until the advent of the familiar Miller [2] and Macintosh [3] laryngoscopes in the 1940s. Tracheal intubation was achieved using one of these direct visualisation devices until Murphy described flexible bronchoscopic tracheal intubation in 1967 (using a choledocoscope [4]), and the lever adaptation of the Macintosh laryngoscope was developed by McCoy in 1993 [5]. Whilst all of these devices were designed to facilitate exposure of the glottis and tracheal tube passage, intubations were not always successful. Upon witnessing the occasional struggles of his anaesthesiology colleague, the surgeon Jack Pacey was prompted to develop the first modern day videolaryngoscope (the GlideScope) [6]—with the principal difference being that a videolaryngoscope achieves an *indirect* view of the glottis, permitting tracheal intubation independent of a direct line of sight. 

Tracheal intubation is predominantly a simple, effective, and safe intervention; however, when primary airway management techniques fail, the situation can rapidly spiral out of control with catastrophic consequences. The Fourth National Audit Project (NAP4) of the Royal College of Anaesthetists (RCoA) described the complications relating to airway management in the UK. It identified that almost half of the incidents reported resulted from problems with tracheal intubation [7]. Repeated laryngoscopy attempts and failed intubation have been shown to be associated with significant patient morbidity, including progression from a safe, controlled situation where oxygenation is possible to the emergency “Can’t Intubate Can’t Oxygenate” (CICO) situation and/or cardiac arrest [8]. 

Videolaryngoscopy (VL) has been shown to improve both first-pass intubation success and to reduce the proportion of difficult glottic views when compared with direct laryngoscopy (DL) [9,10]. Reduced episodes of hypoxia, reduced airway trauma, and better recognition of inadvertent oesophageal intubation have also been demonstrated [9,10]. Grade 1a evidence [11] demonstrating the superior efficacy of VL over DL as a tool to facilitate tracheal intubation has been presented in two Cochrane reviews. Furthermore, the shared view that VL affords may also be associated with enhanced team performance and training for both the trainee laryngoscopist and wider multidisciplinary team (MDT). Increasing recognition of the benefits of VL is reflected in international airway management guidelines—absent from the Difficult Airway Society (DAS) 2004 guidelines [12], familiarity with VL was promoted in the DAS 2015 guidelines [13], evolving to the current recommendations for its *routine* use [14,15,16,17]. 

With increasing exposure to VL, greater familiarity, and the relative ease with which competence can be achieved in asleep patients, VL has emerged as an alternative to flexible bronchoscopic-assisted intubation in select awake patients [18]. 

Despite the relative abundance of evidence supporting VL, DL remains the predominant technique for tracheal intubation in the UK [19] (even in the presence of predictors of difficulty [20]), with current VL usage failing to reflect its now widespread availability [21]. An absence of high-quality evidence and equipment costs have previously been cited as reasons limiting VL acquisition; however, following the upsurge in purchasing during the Coronavirus 2019 (COVID-19) pandemic [22], lack of availability has largely been addressed. Nevertheless, there has not been a commensurate sustained increase in usage, suggesting that there are other factors limiting its widespread adoption and current persistence with DL. 

## 2. Main Body

### 2.1. Epidemiology of Airway Management

Routine tracheal intubation has a high success rate; however, failure can occur—with a reported incidence of approximately 1 in 1–2000 routine cases [23,24]. Inability to intubate the trachea in itself may not lead to significant harm, but concurrent failure to ventilate the lungs or provide adequate oxygenation by other means (“Can’t Intubate Can’t Ventilate”, CICV, or “Can’t Intubate Can’t Oxygenate”, CICO, respectively) can be catastrophic, occurring in approximately 1 in 5–10,000 cases [25,26]. Intubation failure increases dramatically in emergency situations, occurring in approximately 1 in 300–800 cases [23]. Failed intubation can result in pulmonary aspiration, recognised/unrecognised oesophageal intubation, awareness under anaesthesia, airway trauma, hypoventilation and hypoxia resulting in brain injury, adverse cardiac events, and death. CICO may occur as frequently as 1 in 200 in the emergency department (ED) [26] and accounts for over one quarter of all anaesthesia-related deaths [27].

The American Society of Anesthesiologists’ Closed Claims Project (ASACCP) analysis reported that approximately 20% of claims related to respiratory cases (including airway events) [28,29]. Sixty percent of these adverse respiratory events were related to inadequate ventilation, oesophageal intubation, and difficult tracheal intubation.

The NAP4 report identified that almost half of the incidents followed primary problems with tracheal intubation, particularly when a difficult intubation scenario was managed by repeated intubation attempts, deteriorating into a CICV or CICO situation [7]. A suboptimal first attempt at laryngoscopy (leading to failed intubation) is not only unproductive, it can initiate a cycle of progressively diminishing success with every subsequent laryngoscopy attempt [13]. The importance of first-pass success becomes even greater in the critically unwell patient, where multiple attempts at intubation can lead to hypoxia and other life-threatening complications [30]. Compared to intubation achieved on the first or second laryngoscopy attempt, patients requiring further attempts have a seven-fold increase in hypoxia (fourteen-fold increase in severe hypoxia), six-fold increase in oesophageal intubation, seven-fold increase in regurgitation, four-fold increase in pulmonary aspiration, and seven-fold increase in cardiac arrest [8]. Consequently, the DAS unanticipated difficult intubation guidelines (2015) emphasised the importance of “maximising the likelihood of successful intubation at *first* attempt, or failing that, to limit the number and duration of attempts at laryngoscopy, to prevent airway trauma and progression to a CICO situation” [13]. 

In half the ASACCP cases analysed, airway difficulty was predicted but no additional precautions were taken to reduce associated risks. Similarly, when DL is undertaken in patients with “predicted difficult” airways, the rate of airway complications has been shown to be 45 times more likely to occur than in those with “predicted easy” airways [31].

The success and relative safety of tracheal intubation is dependent upon maximising first-pass success, reducing the number of difficult intubations encountered, minimising the number of unsuccessful laryngoscopy attempts, preventing the progression towards CICV/CICO, and matching patients with predictors of difficulty with an appropriate difficult airway management strategy. While VL can fail, it offers superiority over DL in the majority of situations. 

### 2.2. Benefits of VL

The first Cochrane review (2016) showed fewer failed intubations when a videolaryngoscope was used, with an increase in the number of easy laryngeal views and reduced difficult views [9]. Additionally, the review also found that there were fewer failed intubations in patients with an anticipated difficult airway when VL was used. Separately, where DL failed as the primary airway management plan, VL delivered a 92% success rate when used as a rescue device [32].

The second Cochrane review (2022) demonstrated increased first-pass success with all types of VL and fewer hypoxaemic events. Hyperangulated (HA) blades reduced accidental oesophageal intubation rates by two- to three-fold [10]. A reduction in applied force during laryngoscopy [33] has also been demonstrated, and it is associated with reduced mucosal/dental trauma and sore throat [9,34]. 

VL may also improve the non-technical aspects of airway management, including team communication, team performance, and training [35,36] by affording the MDT a shared view of the airway. This allows the laryngoscopist’s assistant and wider MDT to anticipate the requirement for additional equipment, optimise external laryngeal manipulation/cricoid pressure, and better understand the anatomical or pathological difficulty being faced.

VL may offer advantages in particular patient groups. These include patients with unstable/limited movement of the cervical spine [34], patients living with obesity, patients requiring optimal positioning of neuromonitoring electrodes in thyroid surgery [37], and patients requiring a tracheostomy [38] or double-lumen tube [39]. 

It has been suggested that VL should be adopted as a standard of care [40,41]. Evidence supporting this approach includes the large observational study conducted by De Jong et al. [42], which demonstrated a significant increase in the proportion of easy intubations in unselected surgical patients, with a significant reduction in the requirement for rescue techniques, challenging glottic views and laryngoscopist-reported difficulty when VL was used. 

The benefits of VL extend beyond operating theatres, with advantages demonstrated in other clinical environments. Specifically, improved first-pass success and reduced oesophageal intubation have been shown in some studies in the intensive care unit (ICU), ED, and pre-hospital settings [43,44,45]. Other studies have been less convincing or only demonstrated advantages for inexperienced laryngoscopists [46]. More research is required, although it is likely that dedicated training in VL for clinicians working in these areas is fundamental to improving success. 

Overall, the level of evidence supporting the superior effectiveness of VL compared with DL for facilitating tracheal intubation should be considered strong; nevertheless, there are some limitations to the Cochrane reviews and other existing studies in this field. There is significant heterogeneity in terms of trial design, reporting methods, and study populations. Aside from the recent EMMA trial (which included 2092 patients) [47], participant numbers are generally relatively limited—the second Cochrane review includes 222 studies with 26,149 subjects, averaging only ~120 per study (of which, approximately half were in the VL treatment arm). Studies using intubation success as their primary outcome measure require more than 1000 subjects in each arm to effectively demonstrate superiority of one device/technique over another [34]. Difficulty in achieving such sample sizes has led researchers to utilise alternate outcome measures, such as time to intubation [34] or intubation success rate by the novice laryngoscopist or medical student [48]. Similarly, given that difficult intubation is relatively uncommon in the general population, studies have had to find substitutes for actual difficulty, with some choosing to use manikins [49] or simulated difficulty [50]. The limitations of manikins are well recognised [51,52]; therefore, these particular studies must be interpreted with a degree of caution. Finally, there is some concern over the generalisability of results—in applying the findings of one study investigating a particular videolaryngoscope model and assuming that this is representative of all videolaryngoscopes; this is a particular issue when broad comparisons are made between DL and VL [53] as if they were two equally well-defined entities. Airway managers must also understand the general principles of VL alongside the unique features of an individual device to achieve optimal results. 

### 2.3. Principles of (Video) Laryngoscopy

VL utilises micro-cameras, prisms, or fibreoptic bundles (dependent upon the model) embedded within the laryngoscope blade to produce a view of the laryngeal inlet that is independent of line of sight—permitting the laryngoscopist to “see around the corner”. In DL, the optimal view is achieved by a combination of lower cervical spine flexion, upper cervical spine extension, and the manipulation of airway structures—principally by the advancement of the laryngoscope blade’s tip into the vallecula followed by the application of a vertical force in an upward direction, which is designed to lift the epiglottis so that the laryngeal inlet can be visualised. This process is often described as aligning the oral, pharyngeal, and laryngeal axes to achieve line of sight (the “three-axis alignment” theory [54]). The “two curve” theory of DL [55] provides an alternative explanation, where the primary curve refers to the oropharyngeal curve and the secondary curve refers to the pharyngo-glotto-tracheal curve, with the point of inflection of the two curves at the base of the epiglottis. Successful intubation is achieved by alignment of these two curves with line of sight and the trachea. In this model, head extension flattens the primary curve, head lift flattens the secondary curve, the “sniffing” position achieves both, and the upward movement of the laryngoscope completes the flattening of the primary curve [56]. 

VL achieves an *indirect* view of the glottis without aligning oral, pharyngeal, and laryngeal axes or without fully flattening the primary and secondary curves. Understanding this difference is fundamental to performing VL successfully. 

#### 2.3.1. VL Design

Videolaryngoscopes can be subcategorised as follows: channelled (conduited) devices that use an embedded fibreoptic bundle (e.g., Pentax); channelled devices that use a series of lenses, prisms, and mirrors (e.g., Airtraq); or unchannelled devices that use video technology (e.g., C-MAC, GlideScope, and McGrath) (Figure 1). King Vision can be utilised as either a channelled or unchannelled device. Rigid optical stylets (e.g., Bonfils, Shikani, and Bullard) are considered to be a separate class of device and are not discussed here. 

For models using video technology (the most common), the image of the glottis is captured by a camera positioned near the tip of the laryngoscope blade, which is then transmitted to a video screen that is either mounted on the laryngoscope handle or a stand-alone monitor. 

The position of the camera close to the blade tip generally heralds a superior view of the glottis compared with DL (simply due to its proximity to the target) [13,40] but also provides a considerably wider field of view, allowing easier identification of anatomical structures—most evident with a HA blade [40]. The caveat to achieving a superior glottic view is that this does not always translate to easier tube delivery through the vocal cords; and, depth perception can also be more challenging, such that familiarity and practice with locally available devices are essential. 

Channelled videolaryngoscopes consist of a rigid/semi-rigid anatomically curved blade that incorporates a channel through which a tube can be loaded and guided towards the glottis. The tube’s tip can be visualised (on the monitor screen or using an eyepiece) throughout the process of tube delivery to the glottis. If an adequate glottic view is achieved, the tube reliably follows the blade’s trajectory through the glottis. However, if the view of the laryngeal inlet cannot be centralised, the channel may guide the tube away from the vocal cords [19]. Minor adjustments of blade alignment and depth can allow for this; alternatively, a bougie with a coudé tip can be passed through the channel and directed towards the glottis. Channelled devices tend to be slightly bulkier than unchannelled devices, necessitating adequate patient mouth opening; however, they have some potential advantages in a patient with a fixed/unstable cervical spine [57]. 

The unchannelled models of videolaryngoscope can be further subcategorised based upon the blade’s type (Figure 2): First, a Macintosh-style blade that integrates video capability and can be used to perform both direct and indirect laryngoscopy (e.g., McGrath Mac, C-MAC, and GlideScope MAC); secondly, a hyperangulated blade, in which the blade has increased curvature (conformed to the primary curve) designed to further improve glottic visualisation in patients with predicted difficult laryngoscopy (e.g., McGrath X-blade, CMAC-D blade, and GlideScope LoPro), that can only be used for *indirect* laryngoscopy. This additional curvature means that the camera points upward towards the laryngeal vestibule axis. 

The variety of different videolaryngoscope designs makes them suitable for a range of clinical scenarios. Understanding why these variations exist and being able to appropriately select the correct device for the clinical situation will ultimately have a bearing on whether the intubation attempt is successful.

#### 2.3.2. VL Technique

The technique, blade insertion depth, magnitude, and direction of applied forces utilised for VL with a Macintosh-style blade differ significantly from that for a HA blade. Glottic visualisation with Macintosh VL requires flattening of the primary curvature (as in DL) during advancement of the blade’s tip into the vallecula. A HA blade follows the natural anatomical curvature without lifting along the axis of the laryngoscope handle and is inserted less far. Visualisation of the glottis (and surrounding structures) is often superior with a HA blade, but tube delivery may be more challenging without the appropriate airway adjunct/manoeuvres. With Macintosh VL, an airway adjunct may or may not be required to deliver the tube to the glottis, and because some degree of flattening of the primary curve has been undertaken, the most suitable adjunct is often a straight bougie with a coudé tip. In contrast, where a HA blade has been used, the delivery of the tube to the glottis almost always requires an airway adjunct to accommodate the angle of delivery. The most suitable adjunct is often a malleable stylet conformed to the blade shape [58] (Figure 3). Some manufacturers specifically recommend and supply bespoke stylets [59]. A bougie with a coudé tip can be used to facilitate tube delivery, although it may not retain the necessary angulation required when employed with a HA blade [60]. Airway adjunct selection may vary according to the anatomical or pathological challenge being faced, individual airway manager experience and training, and institutional preference. Care should be taken when using any intubating adjunct as these devices can lead to intubation failure and airway trauma when used incorrectly [61].

Devices that accommodate both Macintosh and HA blade designs may confer some advantages in allowing operator selection based on the individual patient and clinical circumstances. 

#### 2.3.3. Video Screen Configuration

Videolaryngoscopes are available with the video screen either handle-mounted (e.g., McGrath, Figure 4) or as a stand-alone monitor (e.g., C-MAC); some manufacturers offer both (e.g., GlideScope Go/GlideScope). A separate monitor provides superior visual quality (optics) and a larger image of the airway that can be more easily shared—which may confer potential benefits in training, team performance, and in the MDT management of critically compromised airways [62]. The view from handle-mounted screens *can* be shared, but it necessitates team re-positioning, although these devices are considerably more portable and well suited to non-operating theatre environments.

### 2.4. Choosing the Right Videolaryngoscope for Your Institution 

Given these differences in model design, blade shape, and screen configuration, it is unsurprising that videolaryngoscopes do not perform equally [63]. Studies comparing the performance of different videolaryngoscopes [64,65,66] have been largely inconclusive, failing to identify clear superiority of one design over another. One particular device may outperform another depending on a particular pathology or clinical scenario. 

In selecting the most suitable videolaryngoscope(s) for a particular institution, a number of different factors must be taken into account. These include the following: establishing exactly what is desired from the device at that particular institution/selected clinical environment; determining whether a model with an integrated screen is preferred to a device with a separate monitor; deciding whether single-use or reusable blades are preferred, taking into account the environmental impact, infection control, and decontamination or waste disposal processes; and, finally, the overall financial costs of the device, monitor (if separate), and any disposables or cleaning equipment. A Medtech innovation briefing from the UK National Institute of Health and Care Excellence (NICE) outlined that the cost of videolaryngoscopes can vary by twelve-fold [67]. 

There is an intuitive argument to support the standardised use of at least the primary videolaryngoscope model across multiple hospital sites within the same training region [68]—to promote institutional learning, avoid confusion of multiple models (with subtle technique differences), enable cost scaling, and the implementation of consistent cleaning protocols [69]. 

Proper training in Macintosh VL and HA blade VL (distinct from one another) is also likely to overcome subtle differences between videolaryngoscope models—minimizing issues in the event of supply chain/procurement shortages that necessitate a temporary switch to an alternate model. 

Moving forwards, institutions must look to the findings of large case series of VL use [69] and comparative studies [70] to inform their decision making on which videolaryngoscope will best suit their needs. 

### 2.5. Recommended Use in Guidelines 

While it may not be currently possible to discern the single best videolaryngoscope, the evidence supporting routine VL use is convincing, leading to its increasing adoption in various international bodies’ guidelines. The DAS 2015 unanticipated difficult intubation guidelines recommended that, “a videolaryngoscope should be immediately available at all times and that all anaesthetists should be trained and skilled in their use” [13]. This was expanded upon in the DAS 2018 guidelines for the management of tracheal intubation in critically ill adults [71], which suggested the specific use of a videolaryngoscope in the presence of a recognised difficult airway or as a rescue strategy when DL has failed. 

More recently, VL has been recommended as the first-line device/technique (wherever feasible) in guidelines for the prevention of unrecognised oesophageal intubation [14], the American Society of Anaesthesiologists Difficult Airway Algorithm, and by the Canadian Airway Focus Group [15,72]. 

### 2.6. Training in (Video) Laryngoscopy

VL can be mastered relatively quickly [73,74], and it can be introduced into clinical practice safely with comparatively little training [75].

Conversely, DL and tracheal intubation is a difficult skill to acquire [76] and maintain. This challenge is compounded when opportunities to practice are limited—which may be the case for non-anaesthetists who may only be required to perform tracheal intubation infrequently and only in emergency situations. It is also recognised that emergency intubation can be more difficult to perform, and it is associated with a lower success rate—particularly if performed by inexperienced laryngoscopists [77]. The occurrence of difficulties and/or failure to successfully intubate the trachea constitutes an important cause of complications, such as airway trauma, oesophageal intubation, hypoxia, and pulmonary aspiration [8]. 

Despite the relative ease of VL skill acquisition, several studies have demonstrated that in order to realise the *full* potential of VL, regular use and proper training are still necessary [69,78,79]. The high first-pass rate attained in the EMMA trial can be attributed to the VL training that was delivered, and the most recent Cochrane review suggested that laryngoscopists with over twenty videolaryngoscope uses were more likely to see a reduction in failed tracheal intubation [10]. Laryngoscopists (and their assistants) should not only have an understanding of how the equipment works but also be able to recognise clinical scenarios where its use may not prove effective [80]. Acquiring this level of expertise requires frequent exposure and regular use—the first instance of use should not be when faced with a predicted difficult tracheal intubation. 

VL offers a number of training benefits over DL. The shared image of the airway allows the trainer to visualise the anatomy in real time while the learner performs laryngoscopy. This allows the trainer to help optimise blade position and advise the learner without having to take over the procedure themselves [81]. Similarly, when airway assistants are learning rapid sequence induction with cricoid pressure, the image on the screen allows the learner and their supervisor to see whether the applied cricoid pressure is displacing or compressing the larynx, making the glottic view worse and/or intubation more difficult [82] and enabling them to adjust their technique and help improve the view. 

By blinding the learner (by turning the video screen away from them), the trainer can also safely teach DL using a Macintosh VL blade. This has important implications for teaching laryngoscopy skills to novice laryngoscopists, with VL providing a superior and safer means of teaching instruction than the supervisor peering over the shoulder of the learner that is attempting to perform DL [81,82,83]. 

### 2.7. Latest Developments

More recently, proficiency with the basics of VL has led to the development of advanced techniques utilising this technology.

#### 2.7.1. Awake VL

In patients with an anticipated difficult airway, the gold-standard approach to airway management has traditionally been an awake flexible bronchoscopic-assisted intubation. This remains the case for a number of specific pathologies; however, greater familiarity and confidence (and availability) in VL for use in asleep patients have led to its recognition as an alternative approach in select awake patients that have recognised predictors of airway management difficulty. 

Following local anaesthetic topicalisation of the airway, a videolaryngoscope can be inserted, and a view of the glottis achieved without significant upward lift (especially if a HA blade is employed). This view can then be used to facilitate awake tracheal intubation prior to induction of anaesthesia [84], or this “awake look” may provide sufficient reassurance to permit induction of anaesthesia, followed by asleep intubation [85].

A number of specific advantages of awake VL have been suggested [18], including reduced time to intubation (with comparable patient experience) [86], the ability to direct the tube through the vocal cords under vision (potentially avoiding impingement that may occur with flexible bronchoscopic techniques), the avoidance of the “cork-in-bottle” phenomenon that can occur with a bronchoscope placed within a critically obstructed airway, the greater range of tube diameters that may be utilised, the ability of the videolaryngoscope blade to create/maintain space within the oropharynx to aid tube passage (bypassing excessive tissue, blood, or secretions), and the superior wider angled view of the upper airway that VL permits. 

#### 2.7.2. Video-Assisted Flexible/Fibreoptic Intubation (VAFI)

VAFI [87] is a two-person intubation technique for use in special circumstances (whether as a planned procedure in the anticipated difficult airway [88] or as a rescue technique [89] performed in either the awake or asleep patient). The videolaryngoscope achieves a superior view of the supraglottis/glottis (than that achieved using a bronchoscope), and the blade creates space for easier tube passage. Tube delivery through the vocal cords is facilitated by a flexible bronchoscope, which itself can be viewed and directed using the video screen. Once the bronchoscope is confirmed within the trachea, the preloaded tube can be railroaded off the scope, with the bevel of the tube adjusted to avoid impingement on the cords using the videolaryngoscope views. 

#### 2.7.3. Suction-Assisted Laryngoscopy Airway Decontamination (SALAD)

SALAD is a technique designed specifically for use with VL in heavily contaminated airways (blood, vomitus, secretions, etc.) [90]. A rigid suction catheter, which conforms to the primary curvature, is advanced into the oropharynx ahead of the blade, with continuous suctioning undertaken (under vision). This allows the blade’s camera to remain free of contamination. Once the blade is advanced into the desired position, the suction catheter can then be used to clear the glottic opening and proximal trachea (under vision) before being placed in the upper oesophagus to suction any further regurgitated material during tube delivery.

### 2.8. COVID Pandemic

Videolaryngoscope purchasing, availability, and use increased globally during the COVID pandemic, following international airway management guidelines that advocated first-line VL use. This recommendation was based upon increasing the physical distance of the laryngoscopist from the airway [17] due to concerns over tracheal intubation being an aerosol-generating procedure (now disproven) [91], emphasising first-pass success, and optimising team performance in critically ill patients with physiologically and logistically difficult airways (given often-stretched resources and unfamiliar team members). This upsurge in VL usage across many countries (including low- and middle-income) was associated with reassuring results in terms of first-pass success and complication rates [92,93]. 

Despite these findings and associated increased VL availability, there has not been the sustained increase in VL usage persisting beyond the pandemic that might have been expected. 

### 2.9. Limitations and Controversies

Despite the recognised benefits of VL, its uptake into standard airway management practice has not been commensurate with its availability. A UK national survey in 2017 [19] demonstrated availability in over 90% of hospitals, yet less than one-third of respondents reported regular videolaryngoscope use, and 10% reported only rare use. Furthermore, in the recent analysis of the UK DAS difficult airway database, DL remained the most commonly reported technique for intubation in patients with predictors of potential difficulty [20]. 

A number of factors may be contributing to this disparity between availability and usage. 

The availability of VL at a particular institution does not necessarily equate to availability across all sites where airway management is undertaken. Only 13% of survey respondents reported videolaryngoscope availability in operating theatres, ICU, and ED [19].

Even where videolaryngoscopes are available across all clinical areas, the method of their introduction is crucial—in 50% of survey respondents, videolaryngoscopes were introduced without any formal structured teaching [19], leading to a lack of confidence and impaired skill acquisition.

The wide range of available videolaryngoscope models also plays a part. In 2017, up to fourteen different videolaryngoscope devices were identified in regular clinical use throughout UK hospitals [19]. As trainees rotate around different hospitals within a training region, variation in equipment and practices is likely to reduce familiarity, impair skill retention, and reduce the transfer of skills to other colleagues. 

In an increasing culture of environmental awareness, the availability of only single-use devices at many hospitals may act as a deterrent to regular use. In hospitals with reusable devices, laborious cleaning protocols can be a significant obstacle to routine VL use. The engagement of operating department practitioners/anaesthetic nurses, who are often responsible for blade decontamination processes, is essential.

There is some concern regarding the possibility of “de-skilling”—potentially losing the ability to perform DL when faced with no alternative [94]. Scenarios commonly cited to support this argument include emergency intubation in a hospital area where there is no videolaryngoscope available, the failure of the videolaryngoscope’s battery/light source, or clinicians’ desire to practice medicine in a low-income country where videolaryngoscopes are presumed not to be available. Concerns over de-skilling have been refuted in a cadaveric study comparing DL and VL in novices [95], which demonstrated that the group trained in VL performed better in both disciplines than those trained in DL alone. It has also been shown in clinical practice that trainee anaesthetists trained solely with VL were not found to be disadvantaged when they rotated around hospitals without this capability [14,41,96]. It is also possible (and indeed advantageous) to teach and practice DL using a Macintosh videolaryngoscope. In the event of practising medicine outwith higher-income countries, clinicians’ DL skills should therefore remain intact (notwithstanding that over 50% of lower-income countries now have VL availability). Promoting first-line VL use does not equate to abolishing DL entirely—DL may succeed occasionally where VL has failed [53] (although with sound VL technique, the potential scenarios in which this might occur would now appear to be vanishingly rare).

Another commonly raised concern is that of the bleeding or severely soiled airway, postulating that the videolaryngoscope camera may become easily obscured leading to impaired glottic views, with the assumption that DL is likely to perform better in such circumstances. However, the research evidence suggests that VL outperforms DL even in severely soiled airways [97]. The aforementioned SALAD technique may also be of benefit in these circumstances. 

Finally, the “seeing yourself fail” phenomenon—where a good glottic view is achieved with VL, but the tube cannot be delivered successfully. There may be a small minority of patients in which this issue cannot be overcome; however, for the most part, appropriate case selection, sound VL technique (following appropriate training [98] and regular practice), the employment of the correct airway adjunct, an appreciation of operator and equipment ergonomics, and the appropriate utilisation of a limited number of optimisation manoeuvres usually result in successful tracheal intubation. This phenomenon may occur in the hands of an experienced direct laryngoscopist who subconsciously attempts to align the three axes [56], thereby hindering tracheal intubation with a HA blade [98]. 

Simple strategies to improve intubation success include the following: maintaining the glottis (target) within the centre of the videolaryngoscope screen; use of an airway adjunct matched to the selected blade; use of a stylet in conjunction with a HA blade and conforming the tube to match the blade shape (Figure 3); holding the proximal end of the stylet (rather than at the mid-point) to increase tip manoeuvrability (Figure 5); advancing the HA blade only as far as is necessary to achieve an adequate glottic view (i.e., blade tip does not need to be in the vallecula); and accepting a lower percentage of glottic opening to facilitate tube passage. 

More advanced strategies include the following: introduction of the tube into the oropharynx at 90 degrees to the blade (at the corner of the mouth) (Figure 6), with retromolar rotation towards the glottis, in order to overcome posterior arytenoid impingement—which can sometimes occur when the tube is advanced in-line with the blade; “reverse loading” of the tube to aid intubation in patients with an anteriorly located larynx—this involves orientating the tube 180 degrees as it is loaded onto the stylet so that when it is advanced off the stylet, the “anatomical” curvature of the tube points posteriorly (Figure 7); and, occasionally, when employing a HA blade, the “Cooper manoeuvre” can be useful to help align the tube tip with the glottis—this involves the vertical lifting of the stylet (and tube) along the axis of the straight section of the stylet. 

There is often inertia for adopting new technology/techniques in any field of medicine. A change in how VL is perceived is necessary to affect a change in airway managers’ behaviour—moving away from occasional videolaryngoscope use by necessity (because the patient’s trachea could not be intubated without it) to regular use by choice because of all the advantages it offers to both the patient and the MDT [99]. 

VL is not the panacea for all airway management. There are situations where VL may be impossible (e.g., reduced patient mouth opening that does not permit blade insertion), or when an alternative primary airway management plan (e.g., flexible bronchoscopy, front of neck airway, etc.) may be more appropriate depending upon the specific patient pathology, anatomical variance, relative physiological (in)stability, attending airway manager expertise, and available equipment. The appreciation of the limitations of VL and thorough knowledge of locally available equipment is crucial in selecting the best airway management strategy for each individual patient. VL can still fail such that contingency planning and the ability to perform core skills such as facemask ventilation and supraglottic airway insertion are fundamental to safe airway management. 

### 2.10. The Future

Evidence of benefits to laryngoscopists themselves is starting to emerge. Since VL allows better maintenance of normal posture and reduced blade lifting forces [100], musculoskeletal problems may be less common—appealing in a specialty known to have significant operator upper limb morbidity [101]. 

VL enables a real-time image/video to be recorded at intubation, which could provide invaluable information to clinicians, informing future interventions (increasing patient safety) and documenting specific pathology. Concerns over how to safely store media files and facilitate sharing with approved parties currently limit the widespread adoption of a recording function that already exists with a number of videolaryngoscope models. With the increasing move towards electronic patient records, it seems unlikely that this recording facility will not be exploited [40]. Indeed, such “digital airway footprints” may become a medicolegal requirement. 

An important area requiring further exploration is the current absence of a reliable reproducible system for recording the glottic view and ease of intubation achieved with VL (since one is not necessarily linked to the other, as with DL). The limitations of the Cormack–Lehane grading system (employed internationally for many years to describe the laryngeal view at DL) when applied to VL are well described [102,103]; however, a suitable, universally accepted alternative has yet to be defined. Attempts have been made, including the Fremantle score [104] and the more recent Video Classification of Intubation (VCI) [105]. 

Continued improvements in screen image quality by device manufacturers and projected reductions in device purchasing (and processing) costs are likely to make VL more appealing to skeptics and more attainable globally (particularly where a lack of device availability remains the principal obstacle to use). 

## 3. Conclusions

VL has the potential to revolutionise airway management and significantly improve patient outcomes and safety. The evidence for its benefits over DL is ever-increasing. Its full adoption into routine clinical practice, as a first-line technique to replace DL, is somewhat lagging behind the burgeoning research evidence, with a number of significant barriers still to be overcome. Improved education and training are not only fundamental in achieving a better awareness of its advantages but also in maximising its efficacy and ensuring its sustained utilisation. The inclusion of VL in the most recent international airway management guidelines in its newly elevated role as a first-line technique is a crucial step in matching device usage with its now widespread availability. However, these strategies must be accompanied by a shift in mindset, where VL should be regarded as a basic standard of care for airway management (based upon the myriad advantages it offers to the patient, laryngoscopist, and MDT), rather than as a necessity, such that the patient’s trachea could not be intubated without it. 

## Figures and Tables

**Figure 1 jpm-13-01327-f001:**
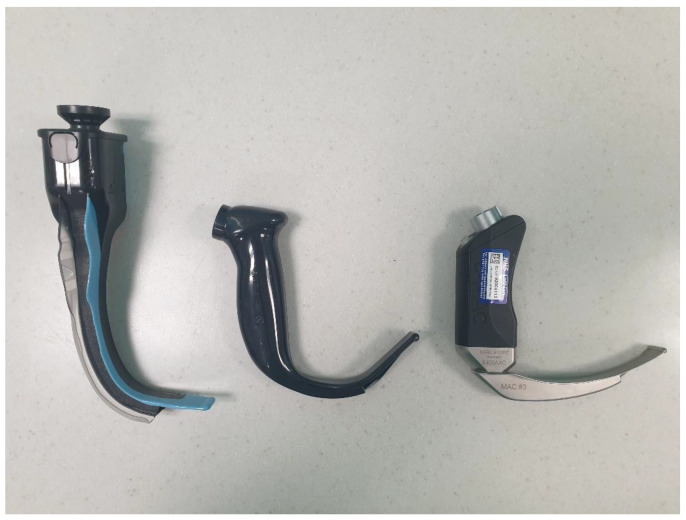
**Videolaryngoscope models.** From **left** to **right**: the Airtraq (channelled device); the GlideScope (unchannelled device; hyperangulated blade shown here); and the CMAC (unchannelled device; Macintosh-style blade shown here).

**Figure 2 jpm-13-01327-f002:**
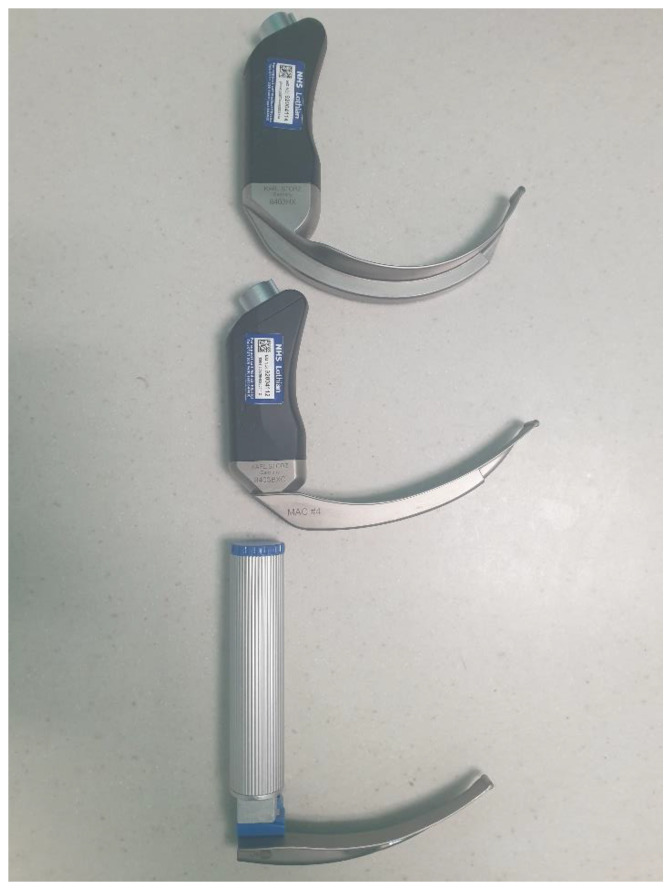
**Videolaryngoscope blade designs.** From **bottom** to **top**: the traditional direct laryngoscope Macintosh blade; the CMAC videolaryngoscope with Macintosh blade; and the CMAC videolarnygoscope with hyperangulated D-blade.

**Figure 3 jpm-13-01327-f003:**
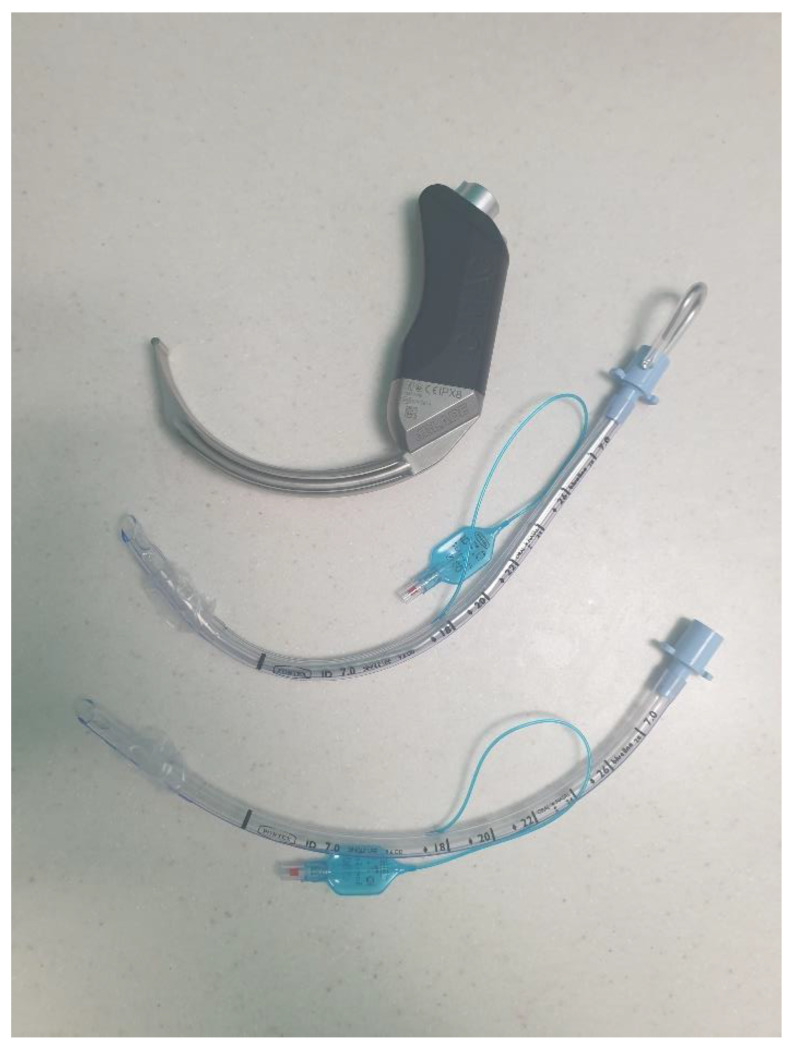
**Airway adjunct conformed to the hyperangulated blade shape.** From **bottom** to **top**: a standard tracheal tube with normal anatomical curvature; a tracheal tube with malleable stylet in situ conformed to the shape of a hyperangulated CMAC-D blade; the CMAC-D videolaryngoscope.

**Figure 4 jpm-13-01327-f004:**
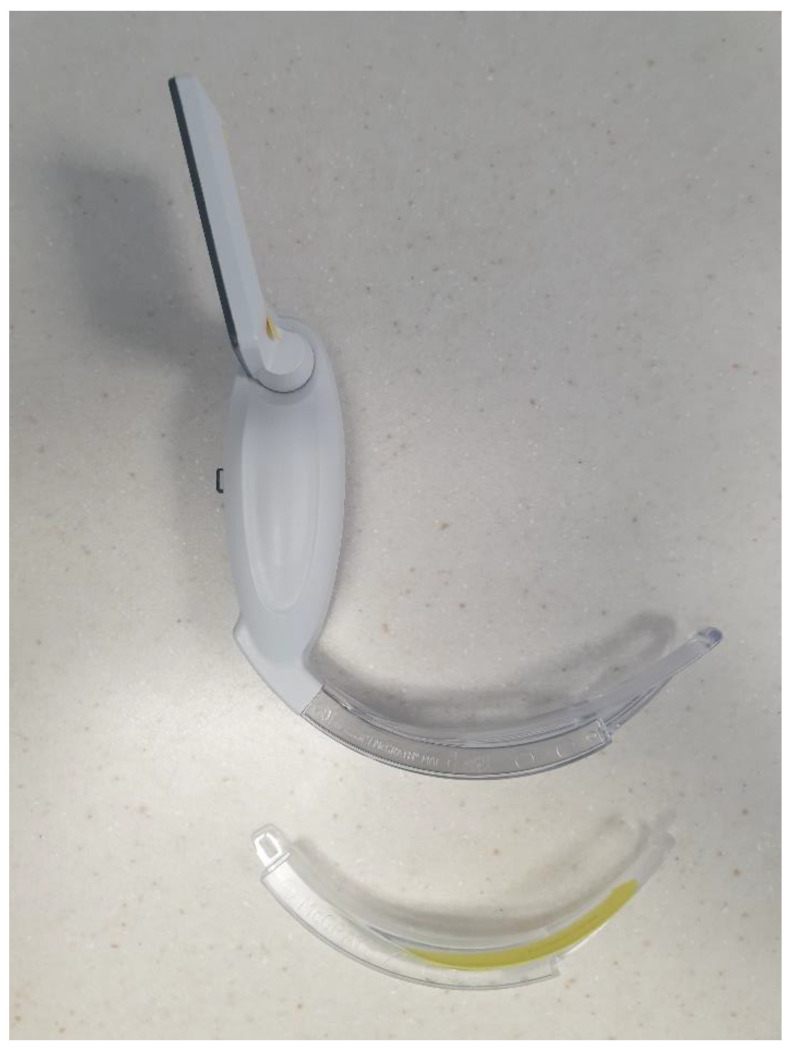
**McGrath videolaryngoscope with handle-mounted screen.** Macintosh blade (attached) and hyperangulated X-blade (unattached).

**Figure 5 jpm-13-01327-f005:**
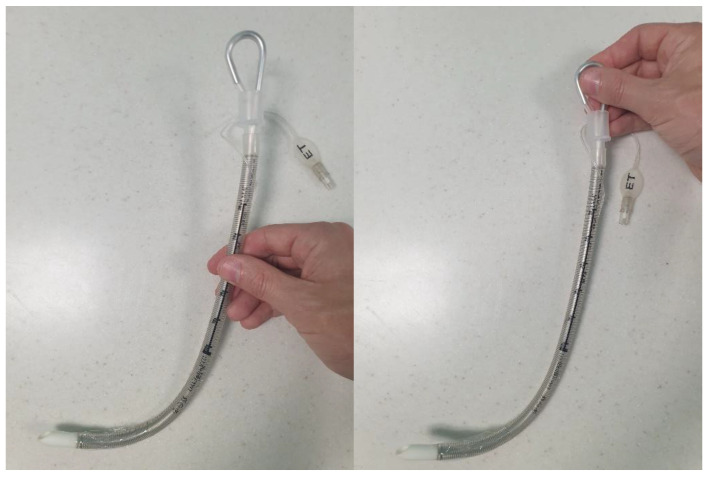
**Hand grip position for manipulating the tracheal tube/malleable stylet when using a videolaryngoscope with a hyperangulated blade. Left** to **right**: hand grip at the mid-point of the tube; hand grip at the proximal end of the tube/stylet (recommended to allow greater manoeuvrability of the tracheal tube tip).

**Figure 6 jpm-13-01327-f006:**
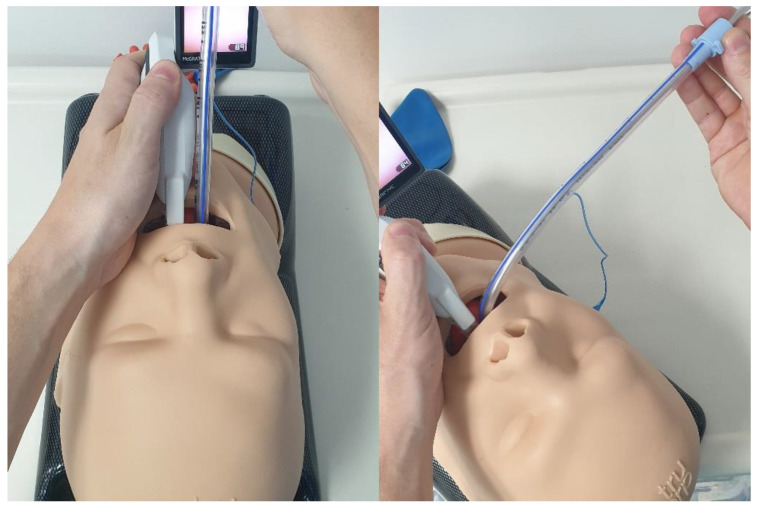
**Technique for the insertion of the tracheal tube/stylet.** From **left** to **right**: in-line (following videolaryngoscope blade); and at 90 degrees to blade (inserted at the corner of mouth, advanced in the retromolar direction and then rotated inwards towards the glottis).

**Figure 7 jpm-13-01327-f007:**
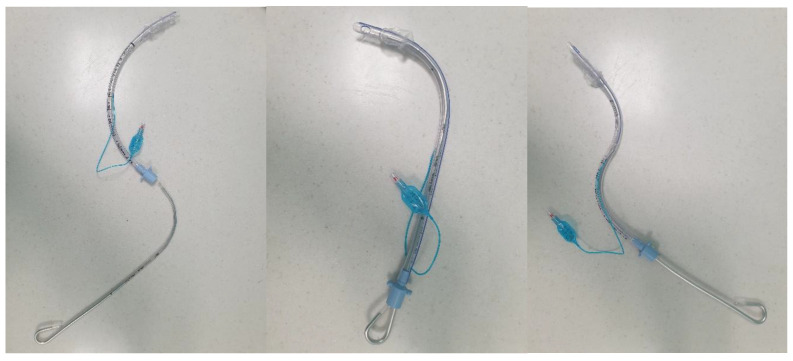
**“Reverse loading” of a tracheal tube onto a curved malleable stylet.** From **left** to **right**: loading of the tracheal tube onto the stylet, with the normal anatomical curvature of the tracheal tube orientated at 180 degrees to the curvature of the stylet; the tracheal tube fully loaded onto the stylet; and the tracheal tube being advanced off the stylet, with the curvature of the tube pointing the tip posteriorly.

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
