# Peer review of "Airway Management: The Current Role of Videolaryngoscopy"

_jpm, 2023, doi:10.3390/jpm13091327_

Round 1

Reviewer 1 Report

Saul and colleagues present a comprehensive review of the literature on videolaryngoscopy. The article is well-written and contains very important information for the anesthesia community.

Although I tried hard, I couldn't find major flaws or misinterpretations of the literature. Congratulations to this well done work!

Author Response

Reviewer 1 summary comments

Saul and colleagues present a comprehensive review of the literature on videolaryngoscopy. The article is well-written and contains very important information for the anesthesia community.

Although I tried hard, I couldn't find major flaws or misinterpretations of the literature. Congratulations to this well done work!

Authors’ reply to reviewer 1 comments

We thank reviewer 1 for their time in reviewing our manuscript and for their kind comments.

Reviewer 2 Report

Very well written, I have enjoyed reading this paper. I thank the editor for giving me this opportunity and I congratulate the authors for compiling this so well. I have small suggestions to add

1. There will be situations where VL can fail such as small jaw, reduced mouth opening, macroglossia, high and anterior larynx , tumours of the larynx, metabolic airway diseases. The readers should not get an impression that VL solves all the problems of difficult intubation. Recognition of these problems in the preoperatively is important to avoid a situation of CICV or CICO. 

2. Personally, I have found use of a flexible bougie and rail roading a tube over better than a metal stylet. This reduces the airway trauma.

3. I agree with authors that not all VL are similar, the quality of picture varies in different products. The future advances could be improvement in the pixel quality and cost reduction

Author Response

Reviewer 2 summary comments

Very well written, I have enjoyed reading this paper. I thank the editor for giving me this opportunity and I congratulate the authors for compiling this so well. I have small suggestions to add

Authors’ reply to reviewer 2 summary comments

We thank reviewer 2 for their time in reviewing our manuscript and for their instructive comments. We have have revised our manuscript accordingly, taking into account their suggestions.

Reviewer 2 comment 1

There will be situations where VL can fail such as small jaw, reduced mouth opening, macroglossia, high and anterior larynx , tumours of the larynx, metabolic airway diseases. The readers should not get an impression that VL solves all the problems of difficult intubation. Recognition of these problems in the preoperatively is important to avoid a situation of CICV or CICO. 

Authors’ reply to reviewer 2 comment 1

In the original manuscript we have stated that VL is not universally applicable or successful:

  • While VL can fail, it offers superiority over DL in the majority of situations. (page 3, line 125)

  • Laryngoscopists (and their assistants) should not just have an understanding of how the equipment works, but also be able to recognise clinical scenarios where its use may not prove effective (page 9, line 352)

We have also added an entirely new paragraph in the “Limitations and controversies” section to acknowledge the limitations of VL more explicitly:

VL is not the panacea for all airway management. There are situations where VL may be impossible (e.g. reduced patient mouth opening that does not permit blade insertion), or when an alternative primary airway management plan (e.g. flexible bronchoscopy, front of neck airway etc.) may be more appropriate, depending upon the specific patient pathology, anatomical variance, relative physiological (in)stability, attending airway manager expertise and available equipment. Appreciation of the limitations of VL and thorough knowledge of locally available equipment is crucial in selecting the best airway management strategy for each individual patient. VL can still fail, such that contingency planning and the ability to perform core skills such as facemask ventilation and supraglottic airway insertion are fundamental to safe airway management.

Reviewer 2 comment 2

Personally, I have found use of a flexible bougie and rail roading a tube over better than a metal stylet. This reduces the airway trauma.

Authors’ response to reviewer 2 comment 2

We recognise that there is variability between individual airway manager’s preferences depending upon level of training and expertise. We have added an additional sentence to reflect the reviewer’s suggestion.

With Macintosh VL, an airway adjunct may or may not be required to deliver the tube to the glottis, and because some degree of flattening of the primary curve has been undertaken, the most suitable adjunct is often a straight bougie with coudé tip. In contrast, where a HA blade has been used, delivery of the tube to the glottis almost always requires an airway adjunct to accommodate the angle of delivery. The most suitable adjunct is often a malleable stylet conformed to the blade shape [58] (Figure 3). Some manufacturers specifically recommend and supply bespoke stylets [59]. A bougie with coudé tip can be used to facilitate tube delivery, though it may not retain the necessary angulation required when employed with a HA blade [60]. Airway adjunct selection may vary according to the anatomical or pathological challenge being faced, individual airway manager experience and training, and institutional preference. Care should be taken when using any intubating adjunct as these devices can lead to intubation failure and airway trauma when used incorrectly [61].

Reviewer 2 comment 3

I agree with authors that not all VL are similar, the quality of picture varies in different products. The future advances could be improvement in the pixel quality and cost reduction

Authors’ response to reviewer 2 comment 3

We agree with the reviewer’s comments regarding future developments and have revised the manuscript accordingly. We have added the following sentence (page 14, line 555):

Continued improvements in screen image quality by device manufacturers and projected reductions in device purchasing (and processing) costs is likely to make VL more appealing to skeptics, and more attainable globally (particularly where lack of device availability remains the principal obstacle to use).